# Development, Objectives and Operation of Return-of-Service Bursary Schemes as an Investment to Build Health Workforce Capacity in South Africa: A Multi-Methods Study

**DOI:** 10.3390/healthcare11212821

**Published:** 2023-10-25

**Authors:** Sikhumbuzo A. Mabunda, Andrea Durbach, Wezile W. Chitha, Paidamoyo Bodzo, Blake Angell, Rohina Joshi

**Affiliations:** 1School of Population Health, University of New South Wales, Sydney, NSW 2052, Australia; rohina.joshi@unsw.edu.au; 2The George Institute for Global Health, University of New South Wales, Sydney, NSW 2042, Australia; bangell@georgeinstitute.org.au; 3Australian Human Rights Institute, University of New South Wales, Sydney, NSW 2052, Australia; a.durbach@unsw.edu.au; 4Health Systems Enablement and Innovation Unit, University of the Witwatersrand, Johannesburg 2000, South Africa; wchitha@hsei.co.za (W.W.C.); pbodzo@hsei.co.za (P.B.); 5The George Institute for Global Health India, New Delhi 110025, India

**Keywords:** health system, health workforce, human resources, health policy, bursary, return of service or RoS

## Abstract

Background: South Africa uses government-funded return-of-service (RoS) schemes to train, recruit and retain skilled health professionals in underserved areas. These educate health professionals locally or internationally in return for a commitment to serve in a specified area for an agreed period. While such schemes are used widely and are funded by substantial public funds, their exact makeup differs across jurisdictions, and little is known about why these differences have emerged or how they influence their effectiveness or impact on the health system. We aimed to fill these gaps through an analysis of the origins, architecture, and evolution of RoS schemes in South Africa. Methods: A multimethod research study including a policy review, a literature review, and semi-structured interviews of policymakers was undertaken between October 2020 and August 2022. The included policy documents and literature were analysed using the Walt and Gilson framework and narrative synthesis. Qualitative data were analysed using inductive, thematic analysis in NVIVO 12. Results: RoS schemes are used as a recruitment and retention strategy and a mechanism to address equity in access to medical education. Whilst there is evidence of RoS schemes existing in 1950, no evidence of beneficiaries was found in databases until 1989. The impact of these schemes is likely being limited by sub-optimal institutional arrangements and poor transparency in their design and implementation. Conclusion: Despite rigorous research methods, the origins of RoS policies in South Africa could not be established due to poor preservation of institutional memory. Opportunities to monitor the value of public investment into RoS programs are being missed and often the underlying objective of the programs has not been well-specified. Policies were found to have been developed and operate in isolation from other health workforce planning activities and thus may not be maximising their impact as a retention and training tool.

## 1. Background

South Africa’s health system is characterised by maldistribution of skilled health professionals [1,2,3,4]. While the country’s ratio of doctors, nurses and midwives of 6 per 1000 population is above the United Nation’s Sustainable Development Goal target of 4.45 per 1000 population, this masks the maldistribution between the private and public health sector, rural and urban areas and a skewed distribution in favour of three of the country’s nine provinces [3,5,6]. To address this maldistribution, South Africa has introduced several strategies including return-of-service (RoS) schemes [3,4]. RoS schemes are recruitment and retention strategies that contract aspiring health professionals and finance their studies in return for obligatory service in underserved areas (generally for the same amount of time that they received funding support) [3,4]. All nine South African provinces have RoS schemes that operate differently and independently of each other. For instance, whilst all nine provinces support a number of South African citizens with programs taken in local institutions, eight provinces send some medical students to Cuba and one province sends some students for a variety of programs to Russia [7,8]. Whilst most of the provincial RoS schemes are administered by the Department of Health, two are partially administered by the Department of Education and two by the Office of the Premier [7]. South African RoS policies have never been systematically evaluated, and there is poor coordination between the various implementing stakeholders, resulting in the failure to retain trained health professionals [7]. Key policymakers have reported the ambiguity of the aims (e.g., preference for rural students, limit of service areas to rural health facilities, internship period not counting towards the service obligation, etc.) of these policies and how this has presented significant challenges in implementation [7]. In this study, we investigate the development, objectives and architecture of these policies through a multimethod approach that is centred on the Walt and Gilson framework for policy analysis [9,10]. An in-depth understanding of how and why the schemes were initiated will help to assess if RoS schemes are meeting these aims and whether they can be used as a reliable health workforce solution, identify their strengths and limitations and facilitate cross-jurisdictional learning.

## 2. Context

Before 1994, South Africa had a non-democratic government, four provinces and ten homelands, which were divided on tribal grounds [11,12]. The health system was equally fragmented. After 1994, the district health system was adopted as the vehicle through which services were delivered using a primary health care approach [13,14]. The public health system was organised into nine provincial health administrations and fifty-two health districts [15]. Provinces are responsible for planning and delivering public healthcare to their population [13,16]. Through a system that has been equated to fiscal federalism, budgets are allocated equitably between provinces from the national treasury to account for over 97% of provincial income [17,18,19,20]. Figure 1 depicts the organisation of South Africa’s health system.

In 2022, South Africa had ten medical schools offering a range of health sciences qualifications such as radiography, dietetics and Bachelor of Medicine and Surgery. After completion, pharmacy and medical graduates have to complete a mandatory internship period of one or two years, respectively [21,22]. This is then followed by a year of compulsory community service, which all other health sciences graduates have to complete [21,22]. Community service is usually considered as part of the obligatory period for RoS beneficiaries.

## 3. Materials and Methods

### 3.1. Design

This study used a multimethod research design with three components: (a) literature review, (b) policy review and (c) semi-structured interviews of policymakers and implementors. Data from these methods were triangulated to understand the historical context, objectives, operation and evolution of the RoS schemes. There was generally good alignment between the three data sources, as such information is presented under a single theme, marked by citations if extracted from the literature or policies and direct statements are in italic font. When there were conflicting findings from the different methods these are highlighted. The checklist for the use and reporting of document analysis (CARDA) and the consolidated criteria for reporting qualitative research (COREQ) were used for the article [23,24].

### 3.2. Recruitment and Data Collection

#### 3.2.1. Literature Review

The relevant literature was identified through a three-step process. First, several electronic databases (Pubmed, CINAHL, CENTRAL, OVID and EBSCO Host) were searched using combinations of the following search terms: (‘bursar*’ OR ‘scholarship*’ OR ‘grant’ OR ‘financial aid’ OR ‘return of service’ OR ‘return for service’) AND (‘health*’) AND (‘South Africa*’). The inclusion criteria pertained to all studies related to the development of policies on the funding of skilled professionals in South Africa but excluded those whose recipients were studying towards a non-health sciences profession or where the word “scholarship” was only restricted to knowledge generation and not funding and those only describing the implementation of the schemes without the policy background. Second, we conducted a manual search of the South African Medical Journal from 1884 (when the publication began) to 31 August 2022, the national archives of the National Department of Education and Health (Pretoria), the South African State Archive (Cape Town and Pretoria) and the University of Cape Town’s (UCT) prospectus (1929–1969), council minutes (1932–1967) and the UCT faculty of health science’s Faculty minutes (1935–1968). Finally, financial assistance handbooks from the UCT and the University of the Witwatersrand (South Africa’s two oldest medical schools) were searched from 1918. We sought help from university librarians and searched state archives. Data were extracted using a designed and piloted Microsoft Excel tool to extract relevant information on the RoS policy context, content, process and actors (Appendix A) [3]. The search and data extraction were conducted between 01 October 2020 and 31 August 2022.

#### 3.2.2. In-Depth Interviews with Key Policymakers

To complement the findings of the literature review, we contacted all nine provincial health offices via email to request access to the policymakers who were in senior executive-level positions with responsibility for the implementation of RoS schemes. A total of 20 policymakers were identified. Participants were excluded if they had less than 2 years of experience in the role or refused to consent (none refused to participate). Interviews were conducted using Microsoft Teams using a semi-structured interview guide (Appendix A) [3,7]. Participants could refer to policy documents during the interview and could participate as an individual (six) or with colleagues (ten). Participants gave consent to audio-recording of interviews using Microsoft Teams and this was supplemented by field notes taken by the interviewer (SAM, a PhD candidate and public health physician). Transcripts were obtained from Microsoft Teams, transferred to Microsoft Word and cleaned by the first author who also translated the non-English language phrases used in the interviews.

#### 3.2.3. Policy Review

Lastly, between 1 October 2020 and 31 May 2022, policymakers were requested to provide all versions of policy documents that were on record since the inception of the RoS schemes including other related documents such as contract templates, correspondence templates, appointment letter templates, beneficiary selection criteria and audit reports. Similar to the literature review, this policy review extracted information on a designed and piloted tool (Appendix A) [3].

### 3.3. Reflexivity

The authorship group includes two former South African RoS beneficiaries, one South African-based health systems researcher who is not an RoS beneficiary, one South African-Australian health and human rights researcher and two Australian health systems researchers who have never benefited from RoS schemes. Whilst all authors have extensive experience in health systems research in LMICs, the first author (SAM) and WWC are South African RoS beneficiaries who were funded for their undergraduate medical studies and have relatives and friends who were RoS beneficiaries. Over the years they have observed the return to service of some RoS beneficiaries and this may influence the interpretations of data and the conclusions drawn from them. To mitigate against the potential bias arising out of this, researchers have engaged with diverse perspectives and critically reflected on their own assumptions throughout the lifespan of the research. The article underwent a rigorous internal peer review process where authors could critique each other until the final version was signed off. The internal criticism was more rigorous due to the diversity of the research group.

## 4. Data Analysis

The included articles and policy documents were analysed using the Walt and Gilson policy analysis framework (Figure 2) [25] and are summarised using narrative synthesis. Qualitative data were analysed using an inductive, manual approach to thematic analysis in NVIVO 12. SAM reviewed and coded all the interviews. BA and RJ reviewed a proportion of the interviews and discussed the themes/codes. The research team met regularly to analyse and interpret the themes emerging from the interviews and group discussions. These discussions helped to refine the emergent themes, make appropriate inferences and synthesise findings. A six-stage approach of familiarisation, coding, theme development, review of themes, theme definition and reporting was followed. Qualitative interview results are presented using two themes from the Walt and Gilson framework (Context and Content combined and Processes and Actors combined), twenty-one primary sub-themes and eight secondary sub-themes that emerged from the data. Analyses were synthesised and reported together under the Walt and Gilson framework and the key themes identified in the interviews. Due to the small size of the population and the political environment under which the participants work, the qualitative data is de-identified and the respondents are assigned random labels not related to their province of origin or names.

## 5. Results

### Document Availability, Archiving and Structure

The results include data from twenty-two publications [8,14,27,28,29,30,31,32,33,34,35,36,37,38,39,40,41,42,43,44,45,46] (eighteen peer-reviewed publications and books, two government legislations, one from UCT council minutes and one from the national library archives), nine policy documents [47,48,49,50,51,52,53,54,55] from six provinces, two national level bilateral agreements [56,57], and interviews from 16 policymakers representing eight provinces (Figure 3). Six of the nine provinces (KwaZulu-Natal (KZN), Limpopo (LP), Mpumalanga (MP), Northern Cape (NC), North West (NW) and Western Cape (WC)) had policy documents, and three of these provinces (KZN, LP and NC) had two different iterations of their documents (one current and one previous) [47,48,49,50,51,52,53,54,55]. Policy documents prior to 2010 were not available. The only country-level RoS policy available is the South Africa-Cuba agreement for the years 2001 and 2012. Appendix A summarises the policy documents [47,48,49,50,51,52,53,54,55,56,57] and Table 1 summarises the rest of the included publications.

## 6. Understanding RoS Schemes in South Africa (Context and Content)

### 6.1. Historical Recognition of Health Worker Shortage and Training Plans

Prior to 1912, sponsorships for the health education of native South Africans were provided by missionary/religious organisations [58,59,60,61,62]. The government’s recognition that around 75% of doctors were in the private sector and largely urban-based (Cape Town, Johannesburg and Durban) and serving a minority of the population led to two Commissions in 1928 (Loram Commission and the Pienaar Commission) [37,38,39,40,41,42,63]. Both Commissions recommended increased funding for the training of black doctors who would serve the rural areas [37,38,39,40,41,42]. However, both reports were largely ignored by the government of the day [37,38,39,40,41,42].

### 6.2. Origins of RoS Schemes

The first identified evidence of government RoS schemes is a 1950 advertisement by the South African Committee on Study and Training, inviting applicants for study in the United States of America (USA) recovered from the UCT council minutes [44]. The next one identified is the 1965 South African Bursary and Scholarship register [43], which advertised three models of state funding for the year 1965 [43]. Figure 4 shows the timeline of South Africa’s RoS schemes based on evidence. Appendix A shows three examples of RoS advertisements from 1965; namely, a bursary loan, a full bursary and a scholarship as the three types of RoS schemes that were available at the time [43].

The provincial policy documents do not provide clarity on when bursary programs were first introduced as none of the founding policies could be found. Besides the KZN province’s acknowledgement of the 2003 policy document as the prior version in its 2010 policy document, no other provincial policy document acknowledged any prior versions [47]. KZN’s 2010 document suggests that the program had “…been functioning for the past decade…”, indicating the presence of the program prior to 2003 [47,48]. The earliest retrievable information is for an individual from Mpumalanga province whose medical studies were funded from January 1989 to December 1995, and a few more were funded in the 1990′s. KZN’s and NW’s earliest retrievable entries are those of beneficiaries who studied in Cuba in 1998, 1999 for Limpopo and 2000 for Northern Cape. We could not retrieve information on beneficiaries funded by the WC province before 2012.

None of the respondents had knowledge of the origins of this policy but confirmed its existence during apartheid (before 1994/pre-democracy):

Participant 1: *Prior to 1994, and prior to the new democracy, our former homelands were affording bursaries for their students. All students were given opportunities, but on a limited scale because those were the homelands. They were catering for their own hospitals… and institutions.*

Participant 2: *…some of the programmes they were there in the previous era (sic)… were just carried over by the new government*.

Instead of a reference to archived documents, the default for respondents was to use prior employment knowledge or the stage at which they were pre-1994 as a reference point to an expectation for them to know of the existence of bursaries before 1994.

Participant 3: *Though, I was still young by then, but I know immediately I went to university as it was during transition in 1994…, I remember very well that there was Kangwane bursary prior 1994 (sic) …I have seen some sisters and brothers who benefited in that programme.*

### 6.3. Types of RoS Schemes Operating in South Africa

The schemes target two types of beneficiaries: internal bursaries aimed at employees of the provincial department of health and external bursaries meant for beneficiaries completing high school [49,50,51,52,53,54]. For both sets, funding support could fund studies either within South Africa or internationally. Only internal bursary recipients have the benefit of part-time study as an option [49,50,51,52,53,54]. Whilst the initial funding for women beneficiaries was 25% less than men in at least some provinces [64]), all provinces now promote gender equity, including in payment levels [43,65].

### 6.4. Justification for State-Funded Educational Initiative Policies

The policies are founded on several legal documents, including the Constitution, Employment Equity Act, Skills Development Act, Public Service Regulations, Public Finance Management Act, National Health Act and Public Finance Management Act [47,48,49,50,51,52,53,54,55]. These mandate the government to make higher education available and accessible to potential recruits through the development and improvement of efficiently managed processes including bursary schemes [47,48,49,50,51,52,53,54,55]. RoS schemes can also be used to diversify the health workforce by affording disadvantaged groups the opportunity to acquire or better their qualifications [47,48,49,50,51,52,53,54,55].

The agreement between South Africa and Cuba has four main objectives: recruitment of medical doctors and lecturers to serve South Africa’s rural and underserved communities; training of South African medical students and post-graduates in Cuba; exploration of possible mutual interests in biotechnology production, pharmaceuticals and scientific research; and any other program that may be agreed upon [56,57].

Despite the lack of conclusive evidence about the origin of RoS schemes, there are several reasons found explaining why they have continued beyond the apartheid era.

(a)Continuity

Some provincial administrations felt compelled to continue with RoS schemes after the democratic transition as they had students already in the system.

(b)Recruitment and retention

Bursaries were seen to respond to critical skills shortages and maldistribution of health professionals, particularly to recruit and retain health professionals in the government sector and rural/remote health facilities. As clearly put by a participant:

Participant 4: *…the objective was actually to address …the prevailing skills deficit in … the province, especially around … your medical doctors who we had a serious shortage [of]…*

The shortage is often noticed when health facilities complain about backlogs or higher patient waiting times, e.g., patients staying long in hospitals due to shortages of orthopaedic surgeons to operate on them, etc.

(c)Social responsiveness to the needs of disadvantaged groups

RoS schemes have played a role in affording future job prospects in the health sector for youth with a socio-economic disadvantage, resulting in a reduction in youth unemployment.

Participant 1: *The objectives of the scheme is …to support and award bursaries to the disadvantaged children… Those whom their parents cannot afford to take them to anywhere (sic).*

(d)Compliance with legislation and government mandates

Fulfilment of governmental imperatives, national development plans, continental agendas and legislation were additional reasons for the maintenance of RoS schemes. Legislation included the Skills Development Act, which mandated employers to set aside some finances to upskill their employees [46]. This capacity building and development is an important pillar in governance. The policies also ensure that the composition of human resources in the health sector reflects the demographic pattern of the general population [14].

Participant 5: *‘… the Employment Equity Act… has an important purpose of promoting employment equity…’ to cater for people living with disability, and to ensure gender and racial balance in the workplace.*

Importantly, the Public Finance Management Act ensures that there is accountability for all government expenditures, including RoS schemes:

Participant 6: *…whatever money that comes out… is accounted for and also to ensure that there is transparency and a fair equitable way in which money is spent…*

(e)Political reasons

The Nelson Mandela Fidel Castro bursary award, also known as a “…national project…” is a result of bilateral agreements between Nelson Mandela and Fidel Castro where previously disadvantaged students from South Africa are sent for medical studies in Cuba. Eight of the nine provinces participate in this program, except for the opposition-governed Western Cape province. Whilst provinces finance the education of the beneficiaries, “…the number is usually decided at the national level, and it is apportioned to the provinces”.

Similarly, the Russian agreement with Mpumalanga is a broader political agreement that the province has with the government of Russia to cooperate on multiple ventures, including human capital development for both youth and government employees. Of the students studying in Russia, about 60% of them are health sciences students.

### 6.5. Who Are RoS Beneficiaries and What Academic Programmes Are Supported by RoS Schemes?

South African citizens and provincial department of health employees are eligible for funding support [47,48,49,50,51,52,53,54,55]. The applicants must be residents of the province, come from a historically disadvantage background aligned with South Africa’s history, have passed matric with excellent results, and meet the admission criteria of the institution (a minimum of 70% for Medicine) [47,48,49,50,51,52,53,54,55]. However, those funded to study medicine in Cuba do not need to have excellent grades in matric [56,57,66]. They are selected by South African provincial health authorities with no input from Cuban universities. They must pass their matric with a minimum of 50% in four core subjects (Physical science, Maths, Biology/life sciences and English) [56,57,66]. Policymakers outlined several other criteria that influenced the awarding of bursaries, including a preference for youth.

Whilst the programs of study are implicitly similar between the provinces, the WC has categorised the funded programs into two (Appendix A) [55]. Approximately 80% of WC bursaries are to be allocated to Health services and 20% to degrees related to support services such as engineering, human resources or health economics [55]. The latter categories also have to serve in the health system as part of RoS schemes [55].

### 6.6. Targeted Recruitment of Beneficiaries Who Are Likely to Address Health Worker Shortages

There is an expectation that beneficiaries will return to serve in under-resourced communities, especially in rural areas and underserved health facilities, as pointed out by a participant:

Participant 1: *People will not want to go to such type of areas, but when we are recruiting, we are looking to say: ‘Ok, let us recruit a student… who will at the end of the year (sic), he will enjoy’. He will have a passion of saying: ‘I am going to serve my people. I am going to serve my relatives. I am going to serve the people that made me to be whom (sic) I am’...*

### 6.7. Noted Shortcomings in Beneficiary Selection

Participants confirmed the presence of loopholes, such as lying about parental income. One participant cited an example of a beneficiary who used their grandmother’s income statement and said that it was their mother’s statement. Due to an inconsistent, random verification process, which is at times conducted by social workers, potential beneficiaries could be deceptive about the proof of residency in the province and/or district.

### 6.8. What Are the Obligations of Beneficiaries?

Beneficiary obligations for external beneficiaries are similar for all the provinces [47,48,49,50,51,52,53,54,55]. Beneficiaries may not enter into any other RoS scheme and must self-fund their studies of a specific module or academic year if they repeat it [47,48,49,50,51,52,53,54,55]. Other than maintaining a good academic record throughout their studies, beneficiaries are obliged to return to their source province for service after completion of their studies.

On the period of funding, the NC province (in both policy documents) and the WC province go beyond requiring beneficiaries to serve for each completed year of funding support and expand to say “…or any part thereof …”, referring to any fraction of a year [53,54,55]. The NC and NW provinces specify that part-time bursary holders will only have to serve for one year after attainment of the qualification. The WC and LP provinces exclude internship, while the MP province’s policymakers also say that internship is excluded; this is contradicted by appointment letters from 2007 to 2010, which state that internship will count towards the service obligation (Appendix A) [49,50,52,53,54,55]. According to participants (including those from MP), pharmacy and medical graduates can do internships anywhere because they are still part of their studies and, therefore, do not count towards the service obligation even if completed within the same province that funded their studies.

LP and MP make provision for being unable to employ beneficiaries due to financial constraints [49,50,51]. In this situation, described to be unlikely, the MP province suggests that such beneficiaries would either be absorbed into a traineeship program or released from the compulsory service obligation [51]. This was further noted in a correspondence template one province sent to beneficiaries in 2020 to release them from the contract (Appendix A).

After completion of their pre-final year (fifth year) in Cuba, students return to the country to complete their studies at a South African university. All other internationally qualified medical practitioners are required to write a board exam before registration. The first cohort of five Russian-trained MP medical graduates completed and were unemployed at the time of the interview as they had to prepare for the Health Professions Council of South Africa (HPCSA) board exams.

### 6.9. Is There Room for Contract Variation?

Beneficiaries who repeat an academic year or component (e.g., Appendix A) do so at their own cost [47,48,49,50,51,52,53,54,55]. KZN, LP and NC do, however, make provisions to vary the contract and accommodate such beneficiaries by adding a year of service obligation in their contract if the motivation is accepted by the accounting officer [47,48,49,50,53,54]. A participant narrates the following:

Participant 7: *My experience with students in an effort to…what word shall I use? Bypass, circumvent, workaround the policy provision that says: ‘repeat at own expense’. Most students would come and make a presentation to the Department*.

Other variations that could be considered include the death of the beneficiary (KZN, NC and WC), ill-health incapacity or disability (KZN, NC and WC), medical specialisation (LP), any postgraduate studies (WC), overseas study and research (WC) and for gaining experience in another province [47,48,49,50,53,54,55].

### 6.10. Funding Model

Except for the WC province, which uses a capped fee model, all the other provinces settle the university invoice to cover 100% of all study-related costs including tuition, prescribed textbooks, medical equipment, meals and accommodation [47,48,49,50,51,52,53,54,55]. The LP province also makes provision for transport to clinical excursions, laptops and computer consumables [49,50]. The key comparisons between provinces are summarised in Appendix A.

Participant 2: *…the department will take care of the student’s needs whilst at the university… even to an extent of even including transportation to and from the university.*

All fees and living expenses are covered for those studying in Cuba, including the translation of travel documents, health insurance, and full costs of their reintegration into a South African university [56,57].

Beneficiaries who fail to fulfil contractual obligations are required to repay as a lumpsum all the monies spent on them for their studies, with interest at a rate determined by the National Minister of Finance [47,48,49,50,51,52,53,54,55].

### 6.11. Use of Policy Development Framework

Policymakers were unaware of the World Health Organization’s (WHO’s) framework for human resources for health in Africa and none of the provinces had used any policy development framework for their policy documents [67,68].

Participant 7: *To be honest with you, I can’t remember an instance where we printed that document or produced that document and said: ‘let’s develop our policy and let’s make sure… here is the HRH (Human Resources for Health) policy or document and let us look at it what it says’ (sic)…we rely on how individuals in the meeting recall certain aspects of that document rather than referring to the document specifically (sic).*

### 6.12. Policy Governance and Administration

Provinces have different methods for administering the bursaries as seen in Appendix A [47,48,49,50,51,52,53,54,55].

### 6.13. Presence of Reasons for Review of Preceding Policy Document in Subsequent Versions

According to KZN’s 2010 policy document, the non-retrievable 2003 policy document was reviewed by policymakers because the obligated service area was not limited to the district of origin. There were three broad reasons for the review of this document; namely, equity and retention (e.g., introduction of compulsory mentorship to ensure student support and retention), administrative purposes (e.g., home visits by officials and/or social workers and the use of Department of Home Affairs’ database to verify the parentage of applicants), and political alignment (i.e., alignment of the bursary program with other provincial social responsiveness initiatives) [47,48].

## 7. Process and Actors for Implementation of RoS Policies in South Africa

### 7.1. How Do RoS Schemes Operate in Practice?

In all the provinces, Human Resource Management (HRM) and Human Resource Development (HRD) components (Figure 5) interact with the program at different stages of the process as outlined in Figure 6. First, Human Resource Planning engages with districts and program managers to identify the skills gaps and service needs. Second, these critical skills needs then translate into training needs that have to be funded by the HRD. Third, the national department of health invites health professionals in their final year to apply for either an internship post (medical doctors and pharmacists) or compulsory community service for all other health professionals, through a system known as the Internship and Community Service Programme (ICSP). Human Resource Management (HRM) is then responsible for contracting them as employees and monitoring them from their community service year onwards. Both Figure 5 and Figure 6 are summaries of triangulated findings (mostly interviews) that detail RoS implementation processes and governance in the provinces.

### 7.2. How Does the Policy Planning and Implementation Process Work in Practice?

The NW province states that bursaries will be granted if the field of study is in line with departmental needs [52]. However, the process of determination of departmental needs is not stated for most of the provinces [47,48,49,50,51,52,53,54,55]. According to all the respondents, needs determination and the process of determining beneficiary numbers are not evidence-based. Instead, they are derived through the triangulation of information from various operational processes, including the employee performance management and development system, audit reports, adverse events reports, quarterly reviews, national strategies like the national human resources for health strategy, health facility turnover and attrition rates, etc.

Participant 4: *So, we will be looking into the number of vacant posts that we are not able to… attract skills in… I guess there was nothing scientific about it. I think it was more of a practice or ritual… to say that: ‘no, every year 60 has to be recruited’. …out of these… districts we would equitably share the number and then 60 for us worked out much more better (sic)… to cover all districts.*

According to a participant, these processes could benefit from being proactive and employing health economists, public health specialists and other researchers in general.

Identified skills needs should be of cadres that are scarce and those that are not readily available in the labour market for traditional recruitment. Regular monitoring and filling of these gaps ensures that you do not have high vacancy rates for skilled health professionals.

Expressing a conflicted optimistic and pessimistic view, a participant goes further to say the following:

Participant 8: *…hopefully you’ll be able to keep a bursary holder longer, although that doesn’t work.*

After this process, training needs are derived and passed on to HRD within health. In the case of Mpumalanga and Northern Cape provinces they have to pass on the training needs requiring external applicants to the department of education. The processes are otherwise similar where the bursary offices “…issue application forms…” for the identified fields of study.

Furthermore, not all provinces outline the process of planning, policy review, advertisement, selection, contracting, monitoring, beneficiary support, recruitment into service, systems in place to track defaulters, tracking of defaulters and alerts for when a beneficiary has completed their service obligation [47,48,49,50,51,52,53,54,55]. Only the WC includes the role of external stakeholders and academic institutions in the selection of beneficiaries [55].

According to the KZN province’s policy documents, health institutions should collaborate with districts to plan and prioritise the identified scarce skills per occupational category [47,48]. WC’s planning process places emphasis on budget compliance, which is reviewed on a three-year cycle [55].

### 7.3. How Are the Opportunities Advertised?

This is summarised in Appendix A.

### 7.4. How Are Beneficiaries Selected?

All provinces require some level of academic achievement and verify that they are South African, financially needy and from the province. All provinces (except Gauteng) prioritise applicants of rural origin. Social workers also aid with the verification of the social status of applicants through home visits [47,48,51]. The WC uses the National Students Financial Aid Scheme criteria to assess financial eligibility and 10% of their beneficiaries will be government employees’ children [55]. Selection of nursing beneficiaries is undertaken separately in all the provinces as they mostly rely on in-house Colleges [55]. The accounting officer approves the final list of beneficiaries in all the provinces and candidates are informed telephonically or by mail.

### 7.5. Where Do Beneficiaries Serve Their Contractual Obligation and How Is this Determination Made?

With the exception of the KZN province, which introduced district-based employment of beneficiaries (who are employed in their district of origin after their studies), all other provinces’ beneficiaries will only know their service district immediately before completion of their community service [47,48,49,50]. The MP province’s beneficiaries should agree to serve the MP provincial government or its public entities in any capacity for which the government regards him/her as suitable [51].

### 7.6. How Are the Databases Organised?

KZN, LP, MP and NW provincial policies mention a database and/or register that must be maintained regularly [47,48,49,50,51,52]. Despite the KZN province’s description of the contents and structure of the database, the databases were all sub-par, not following the process flow of implementation, and not interoperable enough with the government’s payroll system to be able to link funding duration and support, and the service records of beneficiaries.

Participant 8: *… if we were paying those bursaries on a monthly basis…then we would have had a financial trail. You don’t know if they failed or passed.*

Participant 4: *I am sure you must have noticed that most of our information is … ‘stored on Excel’… we are using it as a tool to save our information. Which I feel is a limitation. We can always improve on that.*

### 7.7. How Are Beneficiaries Recruited into Service?

The HPCSA and the South African Pharmacy Council (SAPC) accredit health facility internship posts for medical and pharmacy graduates, respectively, cognisant of funded posts. The national department of health is responsible for the allocation platform, but the letters of employment are issued by the province or facility of employment. The year of compulsory community service follows a similar process, only that the HPCSA and SAPC are not involved. HRP and HRM drive a process of gazetting community service posts in the province and apportion per health facility. All health sciences students need to apply for community service or internship through the ICSP system during their final year of study, regardless of whether they are bursary beneficiaries or not. This is a recently (2016) introduced system:

Participant 9: *…in doctor’s time, when you still used to apply for your internship and community service, we still had the paper-based system.*

During their last six months of internship, medical and pharmacy interns will follow the same process when applying for community service. Applicants are given a month to make five equally weighted choices in at least three different provinces if not a bursary holder or during an internship for all graduates. In an indirect test of conscience, bursary beneficiaries are then given a binary “...yes…” or “…no…” option by the system to choose if they are bursary beneficiaries or not. After the internship, they will not be able to choose allocation in another province if the response is affirmative.

Participant 10: *… the system will tell them: ‘…in this province, … these are the hospitals that you can… that have posts because not all hospitals has (sic) got posts’.*

Bursary holders are also restricted by the system from applying for community service posts at Tertiary and Central hospitals. The list of applicants to each province is then sent to that province:

Participant 9: *…we will do a verification to check that all our… bursary holders have applied. If we find that there are bursary holders that we know is (sic) in final year and they didn’t apply then we will question it obviously, to find out why they are not applying… Then towards the end of September/October we will get a list of allocations which will tell us: ‘these are your allocations for the next year’, say for 2022.*

The allocation of community service posts in each province will prioritise bursary beneficiaries.

Participant 5: *… firstly, we will look at where is the need. Then we look at who was allocated a bursary from which district. …when we allocate, and we are five of us that’s… applied, we will look at… if; are you married, or you’ve got children that’s in school… that will be a special consideration for you. …then we will look at; where… when you were granted the bursary where were you from?*

After completion of community service, the process is inconsistent between the provinces. Non-bursary holders have to either competitively apply for an advertised post and be interviewed or look for a job elsewhere or even go to the private sector. Hospital Chief Executive Officers are told to prioritise bursary beneficiaries in some provinces:

Participant 6: *… ‘these are our bursary holders when you have a vacant post, you have to consider them first to allow them to serve our bursary obligation’.*

Using a list of health facilities with vacant positions, bursary beneficiaries apply for a post by choosing three equally weighted options in one of the provinces. Local facility managers have to confirm that an employee is not a bursary beneficiary before termination on the Personnel and salaries administration system.

### 7.8. What Are the Plans in Place to Strengthen the Policies?

In the KZN province, bursary administrators should be continuously developed to keep abreast with the developments on bursary matters, and they are mandated to analyse the cost implications of the bursary scheme and also calculate the return on investment of the bursary scheme [47,48]. Whilst the LP province’s policy is to be reviewed and influenced by the socio-economic challenges and national priorities and mandates, the NC province’s 2013 policy aimed to review the policy every 12–18 months and the NW province aims to review their 2021 policy document in 2024 (after 3-years) [49,50,52,53,54]. However, the 2020 NC policy document revised this review period to three years after the last publishing date [54]. Other provinces do not state their review dates.

## 8. Discussion

In this study, we found gaps and information asymmetry in the origins of RoS schemes in South Africa between the policy documents (including supporting documents, e.g., contract, funding trail, official correspondence templates, advertisements, etc.), experiences of policymakers, available information in their databases, poor archiving and previously published literature. Some provincial administrations operated their schemes with draft (unsigned) policy documents, which raised audit queries and possibly questioned the enforceability of contractual agreements.

This article has also provided insight into RoS policy implementation in South African provinces. We found a lack of institutional memory of the history of the policies, non-evidence-based decision making and a lack of coordination and monitoring mechanisms. The process of determining the health worker needs of provinces, which is central to the objectives of RoS schemes, is not based on public health needs but instead on historical workforce patterns, thus leaving the inherent assumption that the historical formulation of health facility structures (organograms) was informed by evidence. This lack of evidence could also be one of the reasons for the differences in the schemes between the different South African provinces.

While there were references to the existence of social programs dating back to the 1920s, we could not identify clear evidence of these or what they entailed [37,38,39,40,41,42]. Even though it is clear that there were government RoS schemes present as far back as 1950, the origins are not clear and the founding policy documents could not be located [43]. These schemes have evolved from having a combination of a loan repayment scheme and return-of-service agreement to having only the return-of-service agreement without the requirement that beneficiaries pay back any portion of funding received [3,4,7,43]. Present-day schemes are also more inclusive than some previous schemes that would discriminate based on gender and/or race [7,43,47,48,49,50,51,52,53,54,55,56,57,66].

Policymakers suggest that RoS schemes seem to satisfy their social responsibility role (though this will need quantitative analyses) but are uncertain about their legislative mandate, and the schemes do not seem to fulfil the aim of being a tool used to recruit and retain health professionals in underserved areas. The inability to locate pre-democracy policy documents or even to locate versions of the policy documents that are older than 10 years suggests poor institutional records management systems and information transfer (institutional memory).

The United Nations describes eight characteristics of good governance, two of which are transparency and having a strategic vision [69]. These pillars suggest that good governance needs to be characterised by a free flow of information with processes, institutions and information directly accessible to those concerned with them and supported by clear monitoring mechanisms [69]. On strategic vision, governance systems should be resilient to change in individual role players and should understand the historical, cultural and social complexities [69]. These are unfortunately missing facets with the South African RoS policies as demonstrated by the fact that some policymakers would sometimes attribute the poor archiving to their absence in the department at a particular point. Furthermore, the failure to pre-define the future service area not only questions the validity of the need but also impacts planning and resource allocation.

In 2002, the WHO Regional Committee for Africa adopted a document on policies and plans for human resources for health in Africa [67,68]. This framework was introduced as a foundation for a comprehensive approach to the development and implementation of human resources for health in WHO Africa Member States [67,68]. It is therefore disappointing that none of the participants knew of this framework and none of the RoS policy documents referred to or adopted any parts of the WHO framework [67,68]. The fact that South Africa’s Human Resources for Health 2030 strategy document [17] implicitly applies principles of this framework [67] means knowledge of the framework by national department of health bureaucrats and/or some academics. The weakness could have, therefore, been a combination of failure by WHO Africa to clearly communicate the importance of the framework to member countries and/or the failure of national governments to workshop provincial offices on the framework.

More broadly, the South African RoS policy documents neither follow any framework nor have an evaluation plan to understand the effectiveness of the policy. Other South African Human Resources for Health policy documents like the rural allowance and community service policies also lack structured frameworks for their development and have not been evaluated [70].

Policy analysis is an important tool that independently identifies the need for and relevance of the policy, notes opportunities for advocacy, policy strengths and weaknesses, policy implementation process, policy document review plans and stakeholder involvement [25,71]. Frameworks like the Walt and Gilson framework for policy analysis [25] and the WHO framework [68] not only help with the standardisation of policy documents but also help ensure that important detail is not omitted [9,10,72,73]. The same cannot, however, be said for unstructured approaches as observed in this study where details such as the date of the previous version, date of review, policy objectives, service area, monitoring and evaluation plan etc. were left out of the policy documents. Ambiguities on the inclusion or exclusion of internship in the obligatory period could possibly be avoided as well, further confirming findings of a previous study on South African RoS schemes [7].

Comprehensive policy planning and implementation does not only consider the immediate output (e.g., allocation of bursaries to deserving beneficiaries) but also considers planning for beyond the study period and beyond the period of contracting in service. Planning should also include the undertaking of a thorough risk assessment at the beginning of the implementation phase and review such a strategy at set intervals [74]. This study found instances where this had clearly not been done, including some beneficiaries being given letters of unilateral termination of their contracts due to lack of funds. This is a power imbalance that would be considered a breach of contract if the beneficiaries were the ones writing unilateral letters of termination to the sponsor due to changing circumstances. Furthermore, beneficiaries sent to study Medicine in Russia need to write a Health Professions Council of South Africa board examination to be allowed to practice in the country. Planning for this was not completed on time as the first cohort of five was unemployed for more than 2 years after the completion of their studies.

RoS schemes are meant to strengthen the health system by increasing the pool of skilled health professionals. However, if not properly managed, RoS beneficiaries’ access to specialty training could be delayed due to conditions that could confine them to non-teaching hospitals for a longer time. This could then unintentionally create the same breed of RoS beneficiaries and in a way deny rural and underserved populations the right to access to specialty care closest to where they live. It is therefore commendable to note actions by the Limpopo and Western Cape provincial policies, which accommodate variations of contracts for those intending to further their studies or specialise. Cognisant of the complexities of shortages and maldistribution of health professionals in underserved areas, RoS schemes cannot succeed if they operate in isolation from other existing human resource recruitment and retention policies and programs [1,2,75,76,77,78,79,80,81,82,83,84,85,86,87].

As far as the researchers are aware this is the first study to systematically analyse the history and evolution of government RoS schemes for health professionals in South Africa. The study was limited by the poor preservation of institutional memory within the provincial governments in South Africa and the paucity of the literature on the history of South African RoS schemes. University funding databases were also not included in this study, but that could be a subject of future studies as this study aimed to mainly retrieve the material that government institutions had to implement RoS schemes in South Africa.

## 9. Conclusions

Bursary schemes are an important source of funding for socially deserving future health professionals. Despite an extensive literature review, a thorough policy review and analysis, and interviews with the policymakers and implementers, the origins of these policies in South Africa are not known due to poor preservation of institutional memory and archiving. Opportunities to monitor the value of public investment into RoS programs are being missed, and often the underlying objectives of the programs have not been well-specified. Policies were found to have been developed and operated in isolation from other health workforce planning activities and thus may not be maximising their impact as a retention and training tool. Furthermore, historical structural designs of posts were found to be the main foundation used for future skills needs. Therefore, for RoS schemes to be a reliable and sustainable instrument to redress health professional shortages and maldistribution, planning must be informed by evidence. Such evidence on plans to improve future skills needs in health should be based on population needs and epidemiological, health workforce planning (including health economics) studies. In addition, governments need to invest in strategies to build resilience and strengthen their institutional memory. In this way, institutions will be capacitated to function regardless of the strengths and/or weaknesses of incumbent officials.

## Figures and Tables

**Figure 1 healthcare-11-02821-f001:**
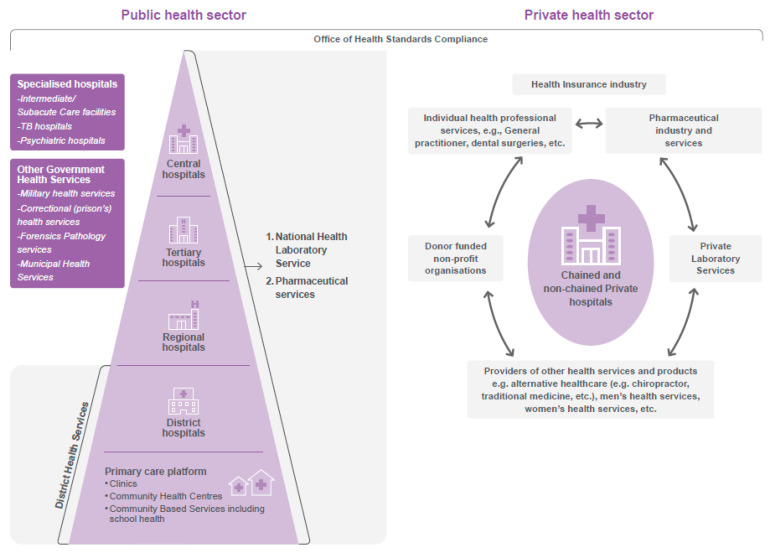
Organisational structure of the South African health system.

**Figure 2 healthcare-11-02821-f002:**
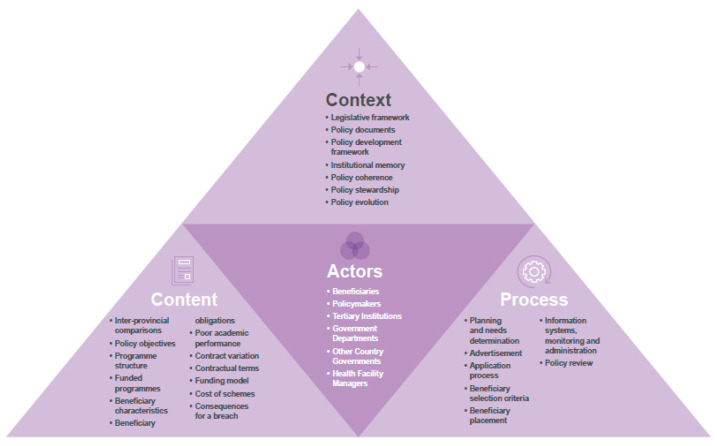
The Walt and Gilson policy framework analysis of South African RoS policies [25,26].

**Figure 3 healthcare-11-02821-f003:**
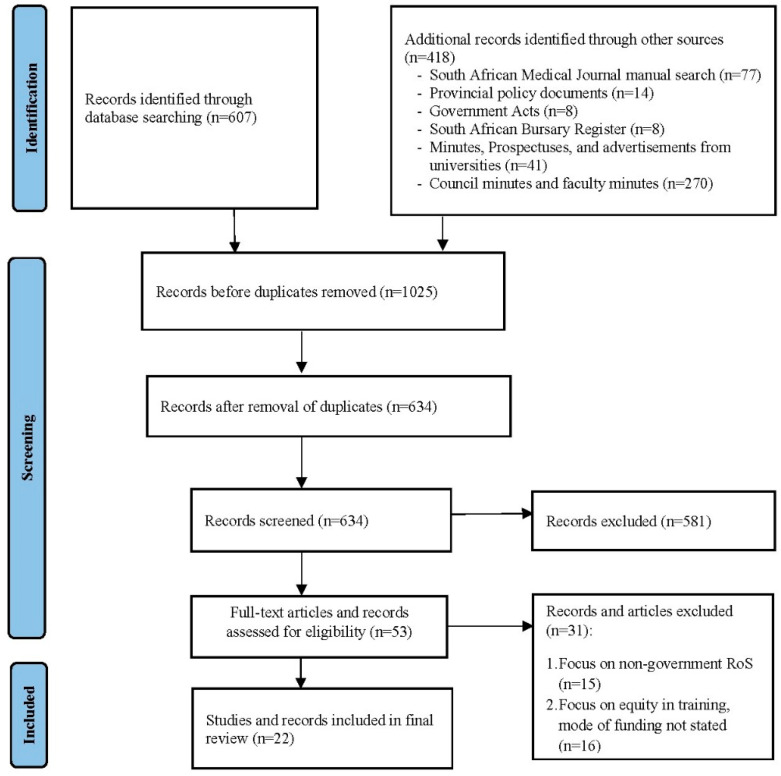
Identification of records via electronic databases and manual searches.

**Figure 4 healthcare-11-02821-f004:**
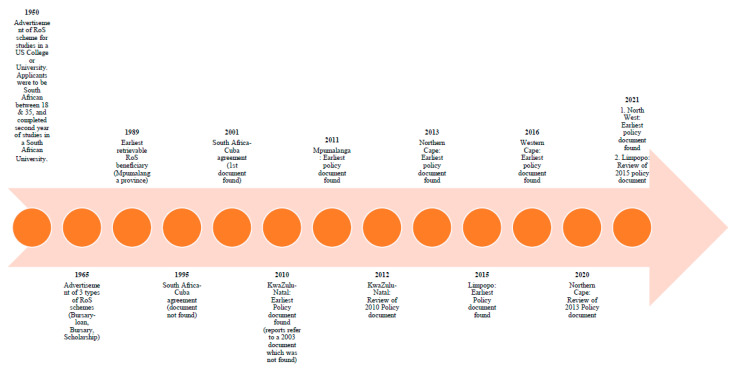
Timeline of South Africa’s RoS policies based on evidence.

**Figure 5 healthcare-11-02821-f005:**
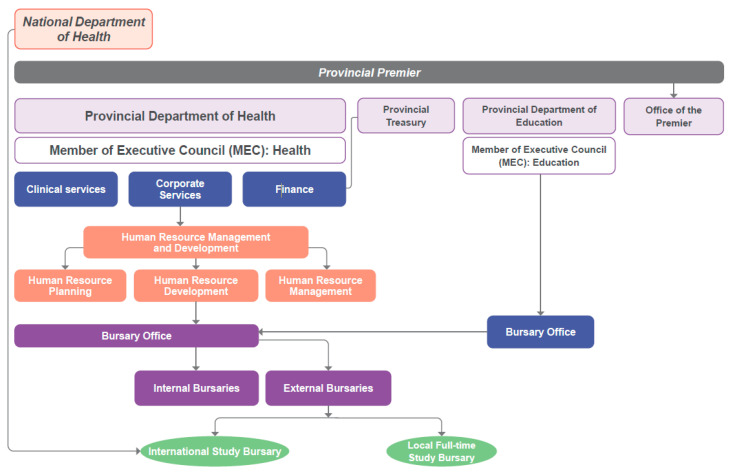
Overview of RoS management in South Africa.

**Figure 6 healthcare-11-02821-f006:**
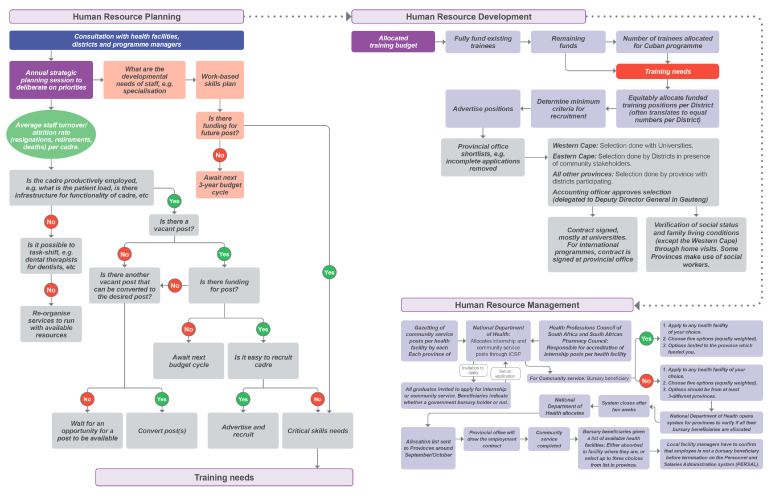
Process of planning and implementing RoS schemes in South Africa.

**Table 1 healthcare-11-02821-t001:** Summary of included publications.

Author(s) Year	Justification for Inclusion
Bateman 2013a [27]	History of South Africa and Cuba bilateral agreement on training medical students.
Bateman 2013b [28]	History of South Africa and Cuba bilateral agreement on training medical students.
Colborn 1991 [35]	Reference to government funding in return for service, for financially distressed medical students at the University of Cape Town.
Delobelle 2013 [37]	History of the South African health system.
Donda et al. 2016 [29]	History of South Africa and Cuba bilateral agreement on training medical students.
Loram 1929 [45]	Proposal for the need to increase intake and funding for Africans who desired to train in Medicine and Public Health.
Motala and Van Wyk 2019 [30]	History of South Africa and Cuba bilateral agreement on training medical students.
Motala and Van Wyk 2021 [31]	History of South Africa and Cuba bilateral agreement on training medical students.
Ncayiyana 2008 [32]	History of South Africa and Cuba bilateral agreement on training medical students.
Quintana et al. 2012 [8]	History of South Africa and Cuba bilateral agreement on training medical students.
Rheinallt Jones and Saffery 1933 [38]	Recommendations for the South African government to increase social assistance to Africans who want to study further, especially in the health sciences.
Rheinallt Jones and Saffery 1934 [39]	Recommendations for the South African government to increase social assistance to Africans who want to study further, especially in the health sciences.
Schapere 1934 [40]	Proposal for the need to increase the training of Africans in health sciences and how this would be funded.
Seekings 2007 [41]	Proposals for increased government social welfare in many sectors including education for both white and black South Africans.
Seekings 2008 [42]	Proposals for increased government social welfare in many sectors including education for both white and black South Africans.
South African Government 1998 [46]	Act on the need to improve knowledge and competencies of government employees, including career pathing and progression (in-service training).
South African government and Loram 1928 [36]	Proposal for the need to increase intake and funding for Africans who desired to train in Medicine and Public Health.
South African National Bureau of Educational and Social Research 1965 [43]	Early advertisement of different types of funding agreements for health science studies, some in return for service.
South African National Department of Health 1997 [14]	White paper proposing the structure of the proposed ‘new South African health system in 1997′ and how the health workforce capacity would be increased.
Squires et al. 2020 [33]	History of South Africa and Cuba bilateral agreement on training medical students.
Sui et al. 2019 [34]	History of South Africa and Cuba bilateral agreement on training medical students.
University of Cape Town 1950 [44]	First advertisement of return-of-service schemes recovered.
Policy documents (9) are embedded within the manuscript in Appendix A.	

## Data Availability

Data used in this study are available from this article.

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
