# Peer review of "Development, Objectives and Operation of Return-of-Service Bursary Schemes as an Investment to Build Health Workforce Capacity in South Africa: A Multi-Methods Study"

_healthcare, 2023, doi:10.3390/healthcare11212821_

Round 1

Reviewer 1 Report

This is a very thorough review of the development and design of health workforce return-of-service schemes in South Africa. It does not in any way attempt to review / evaluate outcomes from these policies. It utilises 3 data sources: (1) review of policy documents (at the province level); (2) literature review of published / public documents; (3) interviews with policymakers. Notably, some of the authors also reported having first-hand prior experience of such RoS policies.

Key concern. The current version of Table 1 (running over 11 pages!) contains a lot of information of different policies across the different provinces; however, its current format makes it near impossible to read and interpret. It looks like it is designed to try and compare between provinces, but this is simply not possible at the moment. With columns this thin, the text of some cells runs over multiple pages and is really hard to read, let alone consider differences / similarities across columns. This may need to be a supplementary file, or perhaps reshape the table so that each province/policy is evaluated one at a time.

Similarly, I’m unclear how helpful Table 3 is in summarising key ‘findings’. It contains information relating to 3 historical (>50 years ago) policies, but don’t greatly contribute to the discussion of findings in the paper.

I didn’t fully grasp either where Figures 5 and 6 were drawn from (Developed for this study? If yes, based on what?), nor what they were trying to demonstrate for the objective of this study. Do these relate wholly to the current process / structure, or are there historical elements within them? If ‘current’, how does that relate to this paper’s focus on the origins of policies, built upon processes up to 50+ years ago. There appears to be minimal crossover of these summaries with other findings of the study (e.g. interviews, literature).

The 2 conclusions in both the Abstract and Main text are substantially different of content and key messages. The Abstract one aligned more in line with the Aim of the paper; I found much of the Main text conclusion went beyond (unsubstantiated) what was found in this paper and needs to be edited.

Author Response

Development, objectives and operation of return-of-service bursary schemes as an investment to build health workforce capacity in South Africa: A multi-methods study

Reviewer 1

Comments and Suggestions for Authors

Comment: This is a very thorough review of the development and design of health workforce return-of-service schemes in South Africa. It does not in any way attempt to review / evaluate outcomes from these policies. It utilises 3 data sources: (1) review of policy documents (at the province level); (2) literature review of published / public documents; (3) interviews with policymakers. Notably, some of the authors also reported having first-hand prior experience of such RoS policies.

Response: Thank you for the comment. Yes, this particular publication does not report on the outcomes and effectiveness of these schemes. We do however, have four other publications that look at the qualitative outcomes, quantitative effectiveness, and the costs of the schemes.

Comment: Key concern. The current version of Table 1 (running over 11 pages!) contains a lot of information of different policies across the different provinces; however, its current format makes it near impossible to read and interpret. It looks like it is designed to try and compare between provinces, but this is simply not possible at the moment. With columns this thin, the text of some cells runs over multiple pages and is really hard to read, let alone consider differences / similarities across columns. This may need to be a supplementary file, or perhaps reshape the table so that each province/policy is evaluated one at a time.

Response: Thank you for the comment. The Table 1 typesetting which was originally in Landscape has been changed by the production. We have heeded the advice and moved it to be a supplementary file (S1).

Comment: Similarly, I’m unclear how helpful Table 3 is in summarising key ‘findings’. It contains information relating to 3 historical (>50 years ago) policies, but don’t greatly contribute to the discussion of findings in the paper.

Response: Thank you for the comment. The Table 3 typesetting which was originally in Landscape has been changed by the production. We moved this Table to be a supplementary file (S2).

Comment: I didn’t fully grasp either where Figures 5 and 6 were drawn from (Developed for this study? If yes, based on what?), nor what they were trying to demonstrate for the objective of this study. Do these relate wholly to the current process / structure, or are there historical elements within them? If ‘current’, how does that relate to this paper’s focus on the origins of policies, built upon processes up to 50+ years ago. There appears to be minimal crossover of these summaries with other findings of the study (e.g. interviews, literature).

Response: Thank you for the comment. These Figures were included to summarise the processes of RoS implementation and governance in the South African provinces. This aligns with the study objectives as we wanted to relate how history would have shaped current RoS schemes. Line 443-445 has now added these words for further clarity:

Both Figure 5 and Figure 6 are summaries of triangulated findings (mostly interviews) that detail RoS implementation processes in the provinces.  

Comment: The 2 conclusions in both the Abstract and Main text are substantially different of content and key messages. The Abstract one aligned more in line with the Aim of the paper; I found much of the Main text conclusion went beyond (unsubstantiated) what was found in this paper and needs to be edited.

Response: Thank you for the comment and advice. The two conclusions have been aligned but keeping in mind that the Conclusion that’s in the Abstract is limited by the Word Count. The Conclusion that’s in the main text also incorporates recommendations in the final four sentences. All points in both conclusions align with the research aims and are stated in the main results. The two Conclusions now read:

Main Text Conclusion:

Bursary schemes are an important source of funding for socially deserving future health professionals. Despite an extensive literature review, a thorough policy review and analysis, and interviews with the policymakers and implementers, the origins of these policies in South Africa are not known due to poor preservation of institutional memory and archiving. Opportunities to monitor the value of public investment into RoS programs are being missed and often the underlying objective of the programs have not been well-specified. Policies were found to have been developed and operate in isolation from other health workforce planning activities and thus may not be maximising their impact as a retention and training tool. Furthermore, historical structural designs of posts were found to be the main foundation used for future skills needs. Therefore, for RoS schemes to be a reliable and sustainable instrument to redress health professional shortages and maldistribution, planning must be informed by evidence. Such evidence on plans to improve future skills needs in health should be based on population needs and epidemiological, health workforce planning (including health economics) studies. In addition, governments need to invest on strategies to build resilience and strengthen their institutional memory. In this way, institutions will be capacitated to function regardless of the strengths and/or weaknesses of incumbent officials.     

Abstract Conclusion:

Despite rigorous research methods, the origins of RoS policies in South Africa could not be established due to poor preservation of institutional memory. Opportunities to monitor the value of public investment into RoS programs are being missed and often the underlying objective of the programs have not been well-specified. Policies were found to have been developed and operate in isolation from other health workforce planning activities and thus may not be maximising their impact as a retention and training tool.

Reviewer 2 Report

Thank you for allowing me to review this paper. I reviewed it with great interest, as it is current and vital. I found topic and issues very interesting and not so well-described previously.

According to the findings from the study, I cannot agree more with the Authors writing  (line 627) "the South African RoS policy documents do not follow any framework, nor do they have an evaluation plan to understand the effectiveness of the policy" and (line 646) "Planning should also include the undertaking of a thorough risk assessment at the beging of the implementations phase and review such a strategy at set intervals". These and several other conclussions I found most important from the study on RoS in South Africa.

I do not have reservations about the manuscript per se, but several recommendations regarding to construction of the text:

1. The text is too long, please be more precise, mainly when the history of RoS is being described.

2. Please include RoS or "return-of-service" into the keywords.

3. Line 176: Walt&Gilson methodology  were cited there, wheras figure 2 with thier framework has invoked to the number 26 from the reference list. 

4. Please cosider changing pages orientation with figures (lines 442 etc.) to make it more visible.

5. Table 1 is so important being extremely non-readable in the same time. Perhaps dividing the table and include in proper sections  (i.e. line 486 and many more) could improve readability. 

Author Response

Development, objectives and operation of return-of-service bursary schemes as an investment to build health workforce capacity in South Africa: A multi-methods study

Reviewer 2

Comments and Suggestions for Authors

Comment: Thank you for allowing me to review this paper. I reviewed it with great interest, as it is current and vital. I found topic and issues very interesting and not so well-described previously.

Response: Noted, thank you for the comment.

Comment: According to the findings from the study, I cannot agree more with the Authors writing  (line 627) "the South African RoS policy documents do not follow any framework, nor do they have an evaluation plan to understand the effectiveness of the policy" and (line 646) "Planning should also include the undertaking of a thorough risk assessment at the beginning of the implementations phase and review such a strategy at set intervals". These and several other conclussions I found most important from the study on RoS in South Africa.

Response: Noted, thank you for the comment.

Comment: I do not have reservations about the manuscript per se, but several recommendations regarding to construction of the text:

Response: Noted, thank you for the comment.

Comment: 1. The text is too long, please be more precise, mainly when the history of RoS is being described.

Response: Thank you for the comment and advice. We do note the length of the article and hope that the edits and the move of Table 1 from the main text will improve its readability. We did not want to assume that all readers understood that the South African provincial structures and health system were different before 1994. We therefore felt that it would be important for readers to be given an overview of the history of South Africa.

Comment: 2. Please include RoS or "return-of-service" into the keywords.

Response: Thank you for the comment. “Return of service” OR “ROS” have been added as keywords.

Comment: 3. Line 176: Walt&Gilson methodology  were cited there, wheras figure 2 with thier framework has invoked to the number 26 from the reference list. 

Response: Thank you for the comment. Walt&Gilson has been added as a reference to Figure 2. Figure 2 is an adaptation of a Figure from another publication (Reference 26).

Comment: 4. Please consider changing pages orientation with figures (lines 442 etc.) to make it more visible.

Response:  Thank you for the comment. I have previously had a conversation with the Journal about something similar and I was assured that the quality and orientation of the Figures will be improved during the final production.

Comment:  5. Table 1 is so important being extremely non-readable in the same time. Perhaps dividing the table and include in proper sections  (i.e. line 486 and many more) could improve readability. 

Response: Thank you for the comment. The Table 1 typesetting which was originally in Landscape has been changed by the production. We have heeded the advice and moved it to be a supplementary file (S1).

Reviewer 3 Report

This is such a great research article. I liked reading and reviewing as I benefit from return-of-service bursary schemes and, more precisely, as an investment to build health workforce capacity. However, I am not from South Africa but from Pakistan. I was a RoS beneficiary for Nursing in Pakistan (from private sector institution). Then MS, leading to PhD in Thailand (from the government of Pakistan), so I can relate to every word of this manuscript personally. 

Overall the manuscript is written very-well. But as I mentioned above I can relate to this research at personal level. So what I was critically looked for and missed is this that how is it possible no single program for nursing was included as you have mentioned in Table 2 and Table3. Though such RoS programs are  primarily designed for Nurses both in developed as well as developing world. 

Second, your main purpose or objective was as you mentioned "...little is known about why these differences have emerged or how they influence their effectiveness or impact on the health system. We aimed to fill these gaps through an analysis of the origins, architecture, and evolution of RoS schemes in South 23 Africa". It seems you didn't achieved the aim of "why these differences have emerged or how they influence their effectiveness or impact on the health system". So, you should revisit, revise and reword this part of the manuscript. Because this is not addressed by this manuscript. Or may be I have missed, in this case, please mentioned where it is. 

For conlsion part, please add future direction. If you couldn't find the required information/data to fill the gap due to poor preservation of institutional memory and archiving. So what needs to be done, to fill this gap. I think the core aim of this research was to fill that gap, which it couldn't. So, the next core contribution of this whole exercise will be to laydown the future direction for what to do next/

Author Response

Development, objectives and operation of return-of-service bursary schemes as an investment to build health workforce capacity in South Africa: A multi-methods study

Reviewer 3

Comments and Suggestions for Authors

Comment: This is such a great research article. I liked reading and reviewing as I benefit from return-of-service bursary schemes and, more precisely, as an investment to build health workforce capacity. However, I am not from South Africa but from Pakistan. I was a RoS beneficiary for Nursing in Pakistan (from private sector institution). Then MS, leading to PhD in Thailand (from the government of Pakistan), so I can relate to every word of this manuscript personally. 

Response: Thank you for the comments, appreciated

Comment: Overall the manuscript is written very-well. But as I mentioned above I can relate to this research at personal level. So what I was critically looked for and missed is this that how is it possible no single program for nursing was included as you have mentioned in Table 2 and Table3. Though such RoS programs are  primarily designed for Nurses both in developed as well as developing world. 

Response: Thank you for the comment. Yes, it’s true that South African nurses are RoS beneficiaries as well but their structure is different from the other programs. Nursing programs are managed from a different division of the department of health and they are considered to be members of staff from their first day of enrolment. They are also enrolled to study within the provincial Colleges (not Universities), therefore making them different from all other programs. This could also explain the possible lack of literature on the nursing programs.

Comment: Second, your main purpose or objective was as you mentioned "...little is known about why these differences have emerged or how they influence their effectiveness or impact on the health system. We aimed to fill these gaps through an analysis of the origins, architecture, and evolution of RoS schemes in South 23 Africa". It seems you didn't achieved the aim of "why these differences have emerged or how they influence their effectiveness or impact on the health system". So, you should revisit, revise and reword this part of the manuscript. Because this is not addressed by this manuscript. Or maybe I have missed, in this case, please mentioned where it is. 

Response: Thank you for the comment, much appreciated. Line 591-593 of the Discussion has added a sentence to express that: “This lack of evidence could also be one of the reasons for the differences in the schemes between the different South African provinces”.

Comment: For conlsion (conclusion) part, please add future direction. If you couldn't find the required information/data to fill the gap due to poor preservation of institutional memory and archiving. So what needs to be done, to fill this gap. I think the core aim of this research was to fill that gap, which it couldn't. So, the next core contribution of this whole exercise will be to laydown the future direction for what to do next/

Response: Thank you for the comment. The Conclusion has been strengthened with three main recommendations (highlighted in green) as follows:

Bursary schemes are an important source of funding for socially deserving future health professionals. Despite an extensive literature review, a thorough policy review and analysis, and interviews with the policymakers and implementers, the origins of these policies in South Africa are not known due to poor preservation of institutional memory and archiving. Opportunities to monitor the value of public investment into RoS programs are being missed and often the underlying objective of the programs have not been well-specified. Policies were found to have been developed and operate in isolation from other health workforce planning activities and thus may not be maximising their impact as a retention and training tool. Furthermore, historical structural designs of posts were found to be the main foundation used for future skills needs. Therefore, for RoS schemes to be a reliable and sustainable instrument to redress health professional shortages and maldistribution, planning must be informed by evidence. Such evidence on plans to improve future skills needs in health should be based on population needs and epidemiological, health workforce planning (including health economics) studies. In addition, governments need to invest on strategies to build resilience and strengthen their institutional memory. In this way, institutions will be capacitated to function regardless of the strengths and/or weaknesses of incumbent officials.     

Round 2

Reviewer 3 Report

The author's have addressed my previous comments very-well. I don't have any further comments. Regards